# Revised NTK Analysis of Optimization and Generalization with Its Extensions to Arbitrary Initialization

## Abstract

Recent theoretical works based on the neural tangent kernel (NTK) have shed light on the optimization and generalization of over-parameterized neural networks, and partially bridge the gap between their practical success and classical learning theory. However, the existing NTK-based analysis has a limitation that the scaling of the initial parameter should decrease with respect to the sample size which is contradictory to the practical initialization scheme. To address this issue, in this paper, we present the revised NTK analysis of optimization and generalization of overparametrized neural networks, which successfully remove the dependency on the sample size of the initialization. Based on our revised analysis, we further extend our theory that allow for arbitrary initialization, not limited to Gaussian initialization. Under our initialization-independent analysis, we propose NTK-based regularizer that can improve the model generalization, thereby illustrating the potential to bridge the theory and practice while also supporting our theory. Our numerical simulations demonstrate that the revised theory indeed can achieve the significantly lower generalization error bound compared to existing error bound. Also importantly, the proposed regularizer also corroborate our theory on the arbitrary initialization with fine-tuning scenario, which takes the first step for NTK theory to be promisingly applied to real-world applications.

## 1 Introduction

Though neural networks (NNs) have achieved great success in practice, it remains a well-known mystery that over-parameterized NNs generalize well and do not suffer from overfitting even with a simple first-order optimization (Neyshabur et al., 2014; Livni et al., 2014; Zhang et al., 2016; Arora et al., 2018), seemingly contradicting the traditional learning theory. To theoretically explain this phenomenon, extensive research has been conducted, and one of the main directions is based on the neural tangent kernel (NTK). Given a NN $f_{\boldsymbol{\theta}}(\cdot)$ parametrized by $\boldsymbol{\theta}$ and $n$ training inputs $\{\mathbf{x}_i\}_{i=1}^{n}$, the NTK is defined as a Gram matrix $\mathbf{H} \in \mathbb{R}^{n \times n}$ induced by the structure of target prediction function whose $(i, j)$-th entry is given by

$$\mathbf{H}_{ij} := \left\langle \frac{\partial f_{\boldsymbol{\theta}}(\mathbf{x}_i)}{\partial \boldsymbol{\theta}}, \frac{\partial f_{\boldsymbol{\theta}}(\mathbf{x}_j)}{\partial \boldsymbol{\theta}} \right\rangle. \tag{1}$$

The NTK was introduced in Jacot et al. (2018) to control the dynamics of learning NNs. In the over-parameterized regime, the trained parameter of NN is close to its initialization, which also makes the NTK almost unchanged throughout the training process. This stability of NTK allows the learning dynamics of NN to be easily analyzed throughout the training process, thus making it possible to derive training and generalization error bounds by using existing learning theory.

As a representative study using NTK, Arora et al. (2019a;b) showed the following important results for over-parameterized NNs:

(a) (Training error bound) A training error bound reflecting a tighter characterization of training speed than that of studies was proposed in Arora et al. (2019a). This bound implies that not only is a network able to represent any finite sample perfectly (as shown in Zhang et al. (2016)), but

the speed at which a network learns training samples varies depending on a complexity measure reflecting how well the data is ordered.

(b) (Generalization error bound) A generalization error bound, referred to as *complexity measure of data* (CMD) was proposed in Arora et al. (2019a). CMD has no stringent conditions on certain properties of the trained NN and the true model; it only depends on input $\mathbf{x}$ and label $y$ of training data and the initial parameter scale $\kappa$ of NN, hence we can compute the bound *before* actually training the network. CMD is considered as one of the most important achievements in the topic of generalizability of over-parameterized NNs and has been the cornerstone of many follow-up studies in recent years (Arora et al., 2019b; Su & Yang, 2019; Oymak et al., 2019; Xu et al., 2019; Zhang et al., 2019; Hu et al., 2019; Du et al., 2018).

The above results (a)∼(b) derived in Arora et al. (2019a;b) provide the upper bounds on the training/generalization error that are *uniformly* available over all network scaling (e.g., initialization) parameter $\kappa$. Note that Arora et al. (2019a;b) focus only on the case where

$$\kappa = o(1) \text{ with respect to } n \tag{2}$$

in order to have meaningful bounds that can converge to zero as $n$ increases.

Surprisingly, however, we prove in this paper that, contrary to the theories of Arora et al. (2019a;b), the training/generalization errors in (a)-(b) *do not* hold when $\kappa$ decreases with respect to $n$ as in Eq. 2. The high-level reason for this is as follows. As trained parameter is known to be close to its initial one in the over-parameterized regime (Jacot et al., 2018; Arora et al., 2019a;b; Du et al., 2018; 2019; Ji & Telgarsky, 2019; Cao & Gu, 2019; Ma et al., 2019; Li & Liang, 2018; Hu et al., 2019; Oymak et al., 2019), the output value of trained NN $|f_\theta(\mathbf{x})|$ is close to that of its initial NN $|f_{\theta(0)}(\mathbf{x})|$. Meanwhile, the output value of its initial NN deceases to zero as $n$ increases if the scale $\kappa$ of initialization decreases with respect to $n$. Thus, it cannot guarantee zero training error under the condition Eq. 2, as the output value of trained NN decreases to zero but the target label does not.

We further resolve the above issue and revise the analyses of Arora et al. (2019a;b) without major modifications of the original statements. Hence, our revision makes it possible for results (a)–(b) to maintain their original meanings and implications without any issue on decreasing $\kappa$. Our revised analyses provide tighter results on training/generalization error bounds, and with these improvements, we can guarantee the bounds to converge to zero even when $\kappa$ is a constant w.r.t. $n$.

Building on the proof technique in the revised theory, we further extend our analysis based on the Gram matrix and network initial parameters drawn from a Gaussian distribution to allow for arbitrary initialization. Grounded on our initialization-independent analysis, we propose the NTK regularizer that can boost the model generalization, which is unavailable in previous studies. Note that our initialization-independent analysis enables NTK theory to be applied to pre-trained networks, which is expected to provide the connections to various practical scenarios such as fine-tuning. Toward this, we empirically verify that our revised theory indeed achieves lower generalization error bounds than the baseline theory and present the potential of the proposed NTK regularizer in fine-tuning scenario.

**Notations.** The sets $\{1, 2, ..., i\}$ and $\{i, i + 1, ..., j\}$ are denoted by $\{i\}$ and $\{i : j\}$, respectively. The Frobenius norm is denoted by $\|\cdot\|$. For a matrix $\mathbf{H}$, $\lambda_{\min}(\mathbf{H})$ denotes the smallest eigenvalue of $\mathbf{H}$. Training samples are given as $n$ input-label pairs $\{(\mathbf{x}_i, y_i)\}_{i=1}^n$ generated from a data distribution $\mathcal{D}(\mathbf{x}, y)$, i.i.d. For simplicity, we assume that $\|\mathbf{x}\| = 1$ for $\mathbf{x} \sim \mathcal{D}$. We denote inputs and labels in the training dataset by $\mathbf{X} = (\mathbf{x}_1, ..., \mathbf{x}_n) \in \mathbb{R}^{d \times n}$ and $\mathbf{y} = (y_1, ..., y_n)^\top \in \mathbb{R}^n$, respectively.

## 2 NTK-BASED ANALYSIS FOR TRAINING/GENERALIZATION ERROR BOUNDS

We first review the training and test error bounds of Arora et al. (2019a) in Section 2.1, and disprove and revise them in Sections 2.2 and 2.3, respectively.

### 2.1 PRELIMINARY: TRAINING/GENERALIZATION ERROR BOUNDS OF ARORA ET AL. (2019A)

Consider a two-layer ReLU network $f_{\mathbf{W},\mathbf{a}}(\mathbf{x})$ with scalar outputs as in Arora et al. (2019a):

$$f_{\mathbf{W},\mathbf{a}}(\mathbf{x}) := \frac{1}{\sqrt{m}} \sum_{r=1}^m a_r \sigma(\mathbf{w}_r^\top \mathbf{x}). \tag{3}$$

Here $\mathbf{x} \in \mathbb{R}^d$ is a given input datapoint, $\mathbf{W} = (\mathbf{w}_1, ..., \mathbf{w}_m) \in \mathbb{R}^{d \times m}$ is the weight parameter in the first layer, $\mathbf{a} = (a_1, ..., a_m)^\mathsf{T} \in \mathbb{R}^m$ is the weight parameter in the second layer, and $\sigma(\cdot)$ is the ReLU activation. The setting indicates that there are $m$ hidden neurons.

Using $n$ samples $(\mathbf{X}, \mathbf{y})$, we train the neural network Eq. 3 so that its prediction function $f_{\mathbf{W}, \mathbf{a}}(\cdot)$ minimizes the following squared error

$$L(\mathbf{W}) = \frac{1}{2} \sum_{i=1}^{n} \left( y_i - f_{\mathbf{W}, \mathbf{a}}(\mathbf{x}_i) \right)^2 \tag{4}$$

by updating the network parameter $\mathbf{W}$ via the discrete time optimization of gradient descent (GD) as

$$\mathbf{W}(k+1) := \mathbf{W}(k) - \eta \frac{\partial L(\mathbf{W})}{\partial \mathbf{W}} |_{\mathbf{W} = \mathbf{W}(k)}.$$

We denote by $\mathbf{u}(k) = (u_1(k), ..., u_n(k))^\mathsf{T} = (f_{\mathbf{W}(k), \mathbf{a}}(\mathbf{x}_1), ..., f_{\mathbf{W}(k), \mathbf{a}}(\mathbf{x}_n))^\mathsf{T} \in \mathbb{R}^n$ the network output with trained parameter $\mathbf{W}(k)$ at $k$-step. The parameter $\mathbf{W}$ is assumed to be randomly initialized as $\mathbf{w}_r \sim \mathcal{N}(0, \kappa^2 \mathbf{I}_d)$ using standard deviation $\kappa$ for $r \in \{m\}$ as in Arora et al. (2019a). Each element of $\mathbf{a}$ is independently initialized (and fixed) as following $\mathcal{U}(\{-1, 1\})$.

By setting the network $f_{\boldsymbol{\theta}}(\mathbf{x})$ and its parameter $\boldsymbol{\theta}$ of NTK in Eq. 1 as $f_{\mathbf{W}, \mathbf{a}}(\mathbf{x})$ and $\mathbf{W}(0)$, respectively, Arora et al. (2019a) derived a specific NTK (with $m = \infty$) as Gram matrix $\mathbf{H}^\infty \in \mathbb{R}^{n \times n}$ as follows: given data matrix $\mathbf{X} = [\mathbf{x}_1, ..., \mathbf{x}_n]$ of $n$ input training samples, $(i, j)$-th entry of $\mathbf{H}^\infty$ is given by

$$\mathbf{H}_{ij}^\infty := \mathbb{E}_{\mathbf{w} \sim \mathcal{N}(\mathbf{0}, \mathbf{I}_d)} \left[ \mathbf{x}_i^\mathsf{T} \mathbf{x}_j \, \mathbb{I}\{\mathbf{w}^\mathsf{T} \mathbf{x}_i \geq 0, \mathbf{w}^\mathsf{T} \mathbf{x}_j \geq 0\} \right] = \frac{\mathbf{x}_i^\mathsf{T} \mathbf{x}_j (\pi - \arccos(\mathbf{x}_i^\mathsf{T} \mathbf{x}_j))}{2\pi}$$

where $\mathbb{I}\{\}$ is the indicator function. We use $\lambda_0$ to denote $\lambda_{\min}(\mathbf{H}^\infty)$. Then, all NTKs obtained by updated parameters $\{\mathbf{W}(k)\}_{k=0}^\infty$ are close to $\mathbf{H}^\infty$ in the over-parameterized regime. Using this fact and extending Du et al. (2018) to hold for *arbitrary* $\kappa$, Arora et al. (2019a) provided the following theorem, which guarantees zero training error with a convergence rate depending on $\lambda_0$.

**Theorem 2.1** (Theorem 3.1 in Arora et al. (2019a)). *Fix a failure probability $\delta \in (0, 1)$. Suppose that $\|\mathbf{y}\| = O(\sqrt{n})$, $m = \Omega \left( \max \left( \frac{n^6}{\lambda_0^4 \kappa^2 \delta^3}, \frac{n^2}{\lambda_0^2} \log \left( \frac{n}{\delta} \right) {}^1 \right) \right)$, $\lambda_0 > 0$, and $\eta = O(\frac{\lambda_0}{n^2})$. Then, with probability at least $1 - \delta$ over the random initialization of $(\mathbf{W}(0), \mathbf{a})$, it follows that for any $\kappa$ and all $k \geq 0$,*

$$\|\mathbf{y} - \mathbf{u}(k+1)\|^2 \leq \left( 1 - \frac{\eta \lambda_0}{2} \right) \|\mathbf{y} - \mathbf{u}(k)\|^2. \tag{5}$$

As a corollary of Theorem 2.1, Arora et al. (2019a) showed a new training error bound reflecting a tighter characterization of training speed such that its convergence rate is mainly affected by the training data belonging to the top eigenspaces of $\mathbf{H}^\infty$. This bound is given as follows.

**Corollary 2.1** (The training error bound, Theorem 4.1 in Arora et al. (2019a)). *Suppose all conditions in Theorem 2.1 hold. Then, with probability at least $1 - \delta$ for $\delta \in (0, 1)$ over the random initialization of $(\mathbf{W}(0), \mathbf{a})$, for all $k \geq 0$,*

$$\frac{1}{\sqrt{n}} \|\mathbf{y} - \mathbf{u}(k)\| = \sqrt{\frac{1}{n} \sum_{i=1}^{n} (1 - \eta \lambda_i)^{2k} (\mathbf{v}_i^\mathsf{T} \mathbf{y})^2} \pm O\left( \frac{\kappa}{\delta} + \frac{n^3}{\sqrt{m} \lambda_0^2 \kappa \delta^2} \right), \tag{6}$$

*where $\{\mathbf{v}_i\}_{i=1}^n$ are orthonormal eigenvectors of $\mathbf{H}^\infty$ and $\{\lambda_i\}_{i=1}^n$ are the corresponding eigenvalues.*

This bound, given as the right-hand side in Eq. 6, reflects the convergence rate in more details by using all the spectral information of $\mathbf{H}^\infty$ (i.e., $\{\lambda_i\}_{i=1}^n$), but the training error bound in Eq. 5 reflects only the least influential part (i.e., $\lambda_0 = \lambda_n$) among these information. This improvement over Theorem 2.1 allows to demonstrate that true labels yield faster learning speeds than random labels (Arora et al., 2019a; Zhang et al., 2016). Meanwhile, for this bound in Eq. 6 to converge to zero, its second term must also decrease to zero and the corresponding condition is given as follows.

**Remark.** For the error term $\frac{\kappa}{\delta}$ in Eq. 6 to decrease to 0 w.r.t. $n$, it should hold that $\kappa = o(1)$ w.r.t. $n$.

Using Theorem 2.1, Arora et al. (2019a) also derived the following generalization error bound, named complexity measure of data (CMD).

**Theorem 2.2** (The generalization error bound, Theorem 5.1 in Arora et al. (2019a)). *Suppose that all conditions except $\lambda_0 > 0$ in Theorem 2.1 hold and we fix a failure probability $\delta \in (0, 1)$. Suppose also that $m = \widetilde{\Omega}(\kappa^{-2} \operatorname{poly}(n, \lambda_0^{-1}, \delta^{-1}))$. Suppose further that $\lambda_0 > 0$ holds with probability at least $1 - \delta/3$ for $n$ i.i.d. training samples $\{(\mathbf{x}_i, y_i)\}_{i=1}^n$ from true model distribution $\mathcal{D}$. Consider any loss function $\ell : \mathbb{R} \times \mathbb{R} \to [0, 1]$ that is 1-Lipschitz in the first argument. Then, with probability at least $1 - \delta$ over the random initialization of $(\mathbf{W}(0), \mathbf{a})$ and the training samples, the neural network $f_{\mathbf{W}(k), \mathbf{a}}(\mathbf{x})$ trained by GD for $k \geq \Omega(\frac{1}{\eta \lambda_0} \log \frac{n}{\delta})$ iterations has population loss[2] $L_{\mathcal{D}}(f_{\mathbf{W}(k), \mathbf{a}}(\mathbf{x})) = \mathbb{E}_{(\mathbf{x}, y) \sim \mathcal{D}}[\ell(f_{\mathbf{W}(k), \mathbf{a}}(\mathbf{x}), y)]$ bounded as*

$$L_{\mathcal{D}}(f_{\mathbf{W}(k), \mathbf{a}}(\mathbf{x})) \leq \underbrace{\sqrt{\frac{2\mathbf{y}^{\mathsf{T}} (\mathbf{H}^{\infty})^{-1} \mathbf{y}}{n}}}_{\text{CMD}} + \underbrace{O\left(\frac{\sqrt{n}\kappa}{\lambda_0 \delta}\right)}_{\text{Error term } \mathcal{E}} + O\left(\sqrt{\frac{\log \frac{n}{\lambda_0 \delta}}{n}}\right). \tag{7}$$

The CMD bound, given in the right-hand side of Eq. 7, only depends on the training samples (e.g., $\mathbf{y}, \mathbf{H}^{\infty}, \lambda_0$) and $\kappa$. This makes it possible to know whether a NN can generalize without actually training the NN, as mentioned above.

**Remark.** For the error term $\mathcal{E} = \frac{\sqrt{n}\kappa}{\lambda_0 \delta}$ in Eq. 7 to decrease to 0 with respect to the sample size $n$, it should hold that $\kappa = o\left(\frac{\lambda_0}{\sqrt{n}}\right)$.

## 2.2 DISPROOF OF EXISTING NTK-BASED TRAINING/GENERALIZATION ERROR BOUNDS

From the remarks above, $\kappa$ should follow $o(1)$ and $o(\lambda_0/\sqrt{n})$ for the training and test errors in Eq. 6 and Eq. 7 to approach zero, respectively. In fact, we have shown in Figure 1 that $\lambda_0$ does not increase with $n$ in standard benchmark datasets, thus $o(\lambda_0/\sqrt{n})$ implies $o(1)$. Hence, $\kappa$ should follow $o(1)$ for both training and generalization errors in Eq. 6 and Eq. 7 to approach zero. These conditions on $\kappa$ can be allowed only if the original Theorem 2.1 is valid for such $\kappa$, as Theorem 2.1 says.

However, in this section, we show that Theorem 2.1 actually does not hold under these conditions on $\kappa$ (i.e., decreasing $\kappa$). Toward this, we visit the case where $\lambda_0$ satisfies the following mild condition

$$\lambda_0 = \mathcal{O}(n^{\gamma}) > 0 \text{ for some constant } \gamma \leq 1. \tag{8}$$

Under Eq. 8, we claim that an additional condition for $\kappa$ (i.e., non-decreasing $\kappa$) is needed for the statements in Theorem 2.1 to hold.

**Theorem 2.3.** *Suppose the condition Eq. 8 holds for a constant $\gamma \leq 1$ and $m = \Omega(n^{3-2\gamma})$. Suppose further $\kappa = o(1)$ for $n$. Then, for any $\eta$ satisfying $0 < \lambda_0 \eta < 2$, there exists a finite integer $k$ (and $n$) with probability at least $1 - \delta$ for $\delta \in (0, 1)$ over the random initialization of $(\mathbf{W}(0), \mathbf{a})$ such that*

$$\|\mathbf{y} - \mathbf{u}(k+1)\|^2 > \left(1 - \frac{\eta \lambda_0}{2}\right) \|\mathbf{y} - \mathbf{u}(k)\|^2. \tag{9}$$

It can be clearly seen that Eq. 5 and Eq. 9 are contradictory and hence the following corollaries hold.

**Corollary 2.2.** *Theorem 2.1 does not hold if the condition $\kappa = o(1)$ w.r.t. $n$ and Eq. 8 hold.*

**Corollary 2.3.** *Corollary 2.1 fails to guarantee that NNs attain zero training loss if Eq. 8 holds.*

**Corollary 2.4.** *Theorem 2.2 fails to guarantee that NNs attain zero gen. err. if $\lambda_0 = O(\sqrt{n}) > 0$.*

The question that naturally arises at this point is how easily the condition Eq. 8 is satisfied in practice. In addition to the observation that $\lambda_0$ does not increase with $n$ in practice as shown in Figure 1, we also find a simple sufficient condition for Eq. 8 to hold, provided in the following proposition:

**Proposition 2.1.** *Suppose that $n$ input samples are not parallel, i.e., $\mathbf{x}_i \neq c\mathbf{x}_j$ for any $c \in \mathbb{R}$ and different $i, j \in \{n\}^2$. Then, $\lambda_0 = O(\sqrt{n}) > 0$ holds.*

Proposition 2.1 confirms that the condition $\lambda_0 = O(\sqrt{n}) > 0$ for Corollary 2.4 holds (i.e., Theorem 2.2 fails) easily in the practically common case where the training data is not parallel.

---

[2]Arora et al. (2019a) claimed that Eq. 7 holds for a general loss, but in fact they implicitly assumed squared loss and did not consider a general loss in the proof.

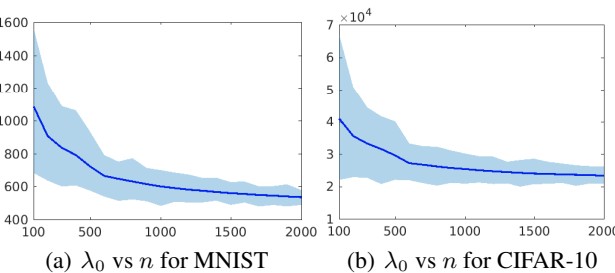

(a) $\lambda_0$ vs $n$ for MNIST  (b) $\lambda_0$ vs $n$ for CIFAR-10

Figure 1: (a) and (b) show the value of $\lambda_0$ w.r.t. sample size $n$ for the standard benchmark image datasets, MNIST and CIFAR-10, respectively. This result shows $\lambda_0 = O(1)$ holds easily in practice.

### 2.3 REVISING NTK-BASED TRAINING/GENERALIZATION ERROR BOUNDS

By Theorem 2.3, $\kappa$ should not decrease w.r.t. $n$ (i.e., $\kappa = o(1)$) in order for the statements in Theorem 2.1 to hold. Thus, we derive tighter bounds so that we avoid the case of setting decreasing $\kappa$ (i.e., $\kappa = \Theta(1)$). In fact, Du et al. (2018) already showed that this is possible for Theorem 2.1 with $\kappa = \Theta(1)$. Here, we revise training and generalization bounds in Corollaries 2.1 and 2.2.

**Theorem 2.4** (Revision of Corollary 2.1). *Suppose all conditions in Theorem 2.1 hold and $\kappa = \Theta(1)$. Then, with probability at least $1 - \delta$ for $\delta \in (0, 1)$ over the random initialization of $(\mathbf{W}(0), \mathbf{a})$, it follows that for all $k \geq 0$,*

$$\frac{1}{\sqrt{n}} \|\mathbf{y} - \mathbf{u}(k)\| = \sqrt{\frac{1}{n} \sum_{i=1}^{n} (1 - \eta\lambda_i)^{2k} \left(\mathbf{v}_i^{\mathsf{T}}(\mathbf{y} - \mathbf{u}(0))\right)^2} + O\left(\frac{n^3}{\sqrt{m}\lambda_0^2\delta^2}\right), \qquad (10)$$

*where $\{\mathbf{v}_i\}_{i=1}^n$ are orthonormal eigenvectors of $\mathbf{H}^\infty$ and $\{\lambda_i\}_{i=1}^n$ are the corresponding eigenvalues.*

Compared to Corollary 2.1, the training error bound in Theorem 2.4 does not have the term $\kappa/\delta$. Accordingly, this tighter bound can converge to zero as the iteration number $k$ increases even in the case for $\kappa = \Theta(1)$. This is formally stated in the following:

**Proposition 2.2.** *Suppose all conditions in Theorem 2.1 hold, $\kappa = \Theta(1)$, and $m = \Omega\left(\frac{n^{6+\alpha}}{\lambda_0^4}\right)$ with any $\alpha > 0$. Then, with probability at least $1 - \delta$ for $\delta \in (0, 1)$ over the random initialization of $(\mathbf{W}(0), \mathbf{a})$, the right hand side of Eq. 10 converges to zero as $k$ and $n$ increase.*

We also revise the CMD bound in Theorem 2.2 as follows, under the condition $\kappa = \Theta(1)$.

**Theorem 2.5** (Revision of Theorem 2.2). *Suppose that all conditions except $\lambda_0 > 0$ in Theorem 2.1 hold and we fix a failure probability $\delta \in (0, 1)$. Suppose also that $\lambda_0 > 0$ holds with probability at least $1 - \delta/3$ for $n$ i.i.d. training samples $\{(\mathbf{x}_i, y_i)\}_{i=1}^n$ from data distribution $\mathcal{D}$, and that $\kappa = \Theta(1)$ and $m = \widetilde{\Omega}(\text{poly}(n, \lambda_0^{-1}, \delta^{-1}))$. Then, with probability at least $1 - \delta$ over the random initialization of $(\mathbf{W}(0), \mathbf{a})$ and the training samples, it follows that for any $k \geq \Omega(\frac{1}{\eta\lambda_0} \log \frac{n}{\delta})$,*

$$\mathbb{E}_{(\mathbf{x}, y) \sim \mathcal{D}} \frac{1}{2} \left| y - f_{\mathbf{W}(k), \mathbf{a}}(\mathbf{x}) \right|^2 = \underbrace{\sqrt{\frac{2(\mathbf{y} - \mathbf{u}(0))^{\mathsf{T}}(\mathbf{H}^\infty)^{-1}(\mathbf{y} - \mathbf{u}(0))}{n}}}_{\text{Revised CMD}} + O\left(\sqrt{\frac{\log \frac{n}{\lambda_0\delta}}{n}}\right). \quad (11)$$

Compared to the original CMD bound in Eq. 7, *the revised version Eq. 11 does not have the error term in Eq. 7,* $(\sqrt{n}\kappa)/(\lambda_0\delta)$, *which is the culprit for generalization error bound to blow up.* By applying Corollary 6.2 in Arora et al. (2019a) to our setting, we can also bound the first term in Eq. 11 even with the introduction of $\mathbf{u}(0)$ exactly as in the original CMD bound:

**Proposition 2.3.** *Suppose $y_i - u_i(0) = g(\mathbf{x}_i) := \sum_j \alpha_j (\beta_j^\top \mathbf{x}_i)^{p_j}$ for all $i \in \{n\}$, where for each $j$, $p_j \in \{1, 2, 4, 6, ...\}$ and $\alpha_j \in \mathbb{R}$ and $\beta_j \in \mathbb{R}^d$ are any constants w.r.t. $n$. Then,*

$$\sqrt{\frac{2(\mathbf{y} - \mathbf{u}(0))^{\mathsf{T}}(\mathbf{H}^\infty)^{-1}(\mathbf{y} - \mathbf{u}(0))}{n}} \leq \frac{6 \sum_j p_j |\alpha_j| \, \|\beta_j\|^{p_j}}{\sqrt{n}} = O\left(\frac{1}{\sqrt{n}}\right). \qquad (12)$$

In that sense, the revised bound directly improves the CMD (in addition to fixing it) by only removing its error term without sacrificing any additional assumption. Also, in Section 4, we will demonstrate that our proposed generalization error bound based on revised CMD could be much smaller than the original bound by showing that revised CMD does not significantly differ in value from the existing CMD for benchmark datasets.

# 3 TOWARDS GENERALIZED ANALYSIS ALLOWING ARBITRARY INITIALIZATION

Although we present the revised NTK analysis in Theorem 2.5, our theory depends on the Gram matrix $\mathbf{H}^\infty$ and the initial network parameter $\mathbf{W}(0)$ (due to the initial network output $\mathbf{u}(0)$), both of which depend on the Gaussian distribution. In this section, we extend our theory that can allow for arbitrary initialization. Based on our extended initialization-independent analysis, we introduce promising applications that can bridge our theory to practice.

## 3.1 PROBLEM SETUP

Similar to our revised NTK theory in Section 2, we could consider 2-layer ReLU networks, but for our theory and the application in the subsequent section (Section 3.3), we consider more generalized settings. Toward this, we consider a multi-layer neural network $f(\cdot)$ with an arbitrary depth, consisting of two components: a feature extractor $\phi_{\mathbf{V}}(\cdot)$ and a linear classifier $f_{\mathbf{W}}(\cdot)$, as follows:

$$h_{\mathbf{V},\mathbf{W}}(\mathbf{x}) := f_{\mathbf{W}}(\phi_{\mathbf{V}}(\mathbf{z})) = f_{\mathbf{W}}(\mathbf{x}). \tag{13}$$

where $\mathbf{z} \in \mathbb{R}^{d_i}$ is an input datapoint, $\mathbf{x} \in \mathbb{R}^{d_h}$ here denotes the representation encoded by the feature extractor $\phi_{\mathbf{V}}(\cdot) : \mathbb{R}^{d_i} \to \mathbb{R}^{d_h}$, and $f_{\mathbf{W}}(\cdot) : \mathbb{R}^{d_h} \to \mathbb{R}^{d_o}$ denotes a (linear) classifier. Each component $\phi(\cdot)$ and $f(\cdot)$ is characterized by the trainable parameters $\mathbf{V}$ and $\mathbf{W}$ respectively. Note that the feature extractor (or also called an encoder) $\phi_{\mathbf{V}}(\cdot)$ can be any neural network that yields $d_h$-dimensional representations, e.g., the feature vectors after global average pooling layer in popular CNNs such as VGG (Simonyan & Zisserman, 2014), ResNet (He et al., 2016), EfficientNet (Tan, 2019), and many other architectures. Throughout the paper, we assume that the parameter of feature extractor is fixed by $\mathbf{V} = \mathbf{V}^*$.

Allowing theoretical analysis in subsequent sections, we replace the linear classifier $f(\cdot)$ with a two-layer ReLU classifier with the vector-valued outputs, i.e., $f_{\mathbf{W}}(\mathbf{x}) = (f_{\mathbf{W}}(\mathbf{x})[1], \cdots, f_{\mathbf{W}}(\mathbf{x})[d_o])^{\mathsf{T}} \in \mathbb{R}^{d_o}$ whose $i$-th output in this setting is computed by

$$f_{\mathbf{W}}(\mathbf{x})[i] := \frac{\sqrt{d_o}}{\sqrt{m}} \sum_{r=1}^{m} \mathbf{a}_i[r] \sigma(\mathbf{w}_r^\top \mathbf{x}), \tag{14}$$

where $\mathbf{x} = \phi_{\mathbf{V}}(\mathbf{z}) \in \mathbb{R}^{d_h}$ is a latent feature, $\mathbf{W} = [\mathbf{w}_1, \cdots, \mathbf{w}_m] \in \mathbb{R}^{d_h \times m}$ is the trainable parameter in the first layer of the replaced classifier, $\mathbf{A} = [\mathbf{a}_1, ..., \mathbf{a}_{d_o}] \in \mathbb{R}^{m \times d_o}$ is the (randomly fixed) weight matrix in the second layer, and $\sigma(\cdot)$ is the ReLU activation.

Similar to the network considered in Section 2, to initialize the parameter $\mathbf{A}$, we directly extend the setting of Arora et al. (2019a) so that only diagonal blocks are randomly initialized as follows: $\mathbf{a}_i[r] \sim \mathcal{U}(\{-1, 1\})$ for $i \in \{d_o\}$ and $r \in \{\overline{m} \cdot i - \overline{m} + 1 : \overline{m} \cdot i\}$, otherwise $\mathbf{a}_i[r] = 0$, where $\overline{m} = m/d_o$ and we assume $\overline{m}$ is an integer throughout the paper for simplicity. In Eq. 14, we use the scaling factor of $\sqrt{d_o/m}$, which is comparable to the scaling of $1/\sqrt{m}$ used in related studies (Bai & Lee, 2019; Nitanda & Suzuki, 2019; Zhang et al., 2019; Du et al., 2018; Arora et al., 2019a; Du et al., 2019). While we have the additional $\sqrt{d_o}$ term, the vector-valued network we consider in Eq. 14 can be divided into $d_o$ scalar-valued networks, and hence the effective number of hidden units for each component is equal to $m/d_o$. We provide the detailed proof on the equivalence between the vector-valued network and the multiple number of scalar-valued networks in Appendix C.

## 3.2 GENERALIZATION ERROR WITH ARBITRARY INITIALIZATION

Let $\mathbf{X} := [\mathbf{x}_1 := \phi_{\mathbf{V}^*}(\mathbf{z}_1), \cdots, \mathbf{x}_n := \phi_{\mathbf{V}^*}(\mathbf{z}_n)]^{\mathsf{T}} \in \mathbb{R}^{n \times d_h}$ be a set of $d_h$-dimensional latent feature vectors obtained by the feature extractor $\phi_{\mathbf{V}^*}(\mathbf{Z})$ with the input dataset $\mathbf{Z} = [\mathbf{z}_1, \mathbf{z}_2, \cdots, \mathbf{z}_n]$.

With the transformed dataset $\mathbf{X}$, we specify the training process by using gradient descent (GD) optimization and assuming the training loss $\mathcal{L}(\cdot)$ as the mean square error (MSE), specified as

$$\mathcal{L}\left(h_{\mathbf{V}^*,\mathbf{W}}(\mathbf{Z}),\mathbf{Y}\right) = L(\mathbf{W}) := \frac{1}{2}\|\mathbf{Y} - F(\mathbf{W})\|^2 \tag{15}$$

where $\mathbf{Y} \in \mathbb{R}^{d_o \times n}$ is a multi-class labels and $F(\mathbf{W}) := [f_{\mathbf{W}}(\mathbf{x}_1),\cdots,f_{\mathbf{W}}(\mathbf{x}_n)] \in \mathbb{R}^{d_o \times n}$ denotes an another representation of prediction function $h_{\mathbf{V}^*,\mathbf{W}}(\mathbf{z})$. Then, the network parameter $\mathbf{W}$ is assumed to be updated via GD on the loss $L(\mathbf{W})$. For our analysis, we define the Gram matrix for each class $i \in \{1 : d_o\}$ computed on the model parameter $\mathbf{W}(k)$ as

$$[\mathbf{H}_i(k)]_{pq} := [\mathbf{H}_i(\mathbf{W}(k))]_{pq} := \frac{d_o}{m}\mathbf{x}_p^\mathsf{T}\mathbf{x}_q \sum_{r \in \mathcal{M}_i} \left[\mathbb{1}\{\mathbf{w}_r(k)^\mathsf{T}\mathbf{x}_p \geq 0, \mathbf{w}_r(k)^\mathsf{T}\mathbf{x}_q \geq 0\}\right]. \tag{16}$$

Note that we use $\lambda_0^i$ to denote $\lambda_{\min}(\mathbf{H}_i(0))$ and $\lambda_0$ to denote $\min(\{\lambda_0^i\}_{i=1}^{d_o})$.

Using the above notations, we present mild conditions required for our theorem to be satisfied.

**Condition 1** (Variable $R$ decreases fast enough for increasing $n$). *Given $\mathbf{W}(0)$, there exists a $R$ satisfying the following condition for each $c \in \{1 : d_o\}$*

$$\frac{1}{n\overline{m}}\sum_{p \in \{1:n\}}\sum_{r \in \mathcal{M}_c}\mathbb{1}\left\{\left|\mathbf{w}_r(0)^\mathsf{T}\mathbf{x}_p\right| \leq R\right\} = \mathcal{O}\left(\frac{\lambda_0}{n^2}\right), \tag{17}$$

*where $\overline{m} = m/d_o$ and $\mathcal{M}_c = \{\overline{m} \cdot c - \overline{m} + 1 : \overline{m} \cdot c\}$.*

**Condition 2** ($\mathbf{W}(0)$ is bounded and $m$ is sufficiently large). *The initial weight $\mathbf{W}(0)$ satisfies the following two conditions*

$$\frac{1}{n\overline{m}}\sum_{p \in \{1:n\}}\sum_{r \in \mathcal{M}_i}\mathbb{1}\left\{\left|\mathbf{w}_r(0)^\top\mathbf{x}_p\right| = \mathcal{O}\left(\frac{n}{\sqrt{m}\lambda_0}\right)\right\} = \mathcal{O}\left(\frac{\min(\lambda_0^2,\lambda_0^3)}{n^4}\right),$$

$$\frac{1}{n\overline{m}}\sum_{p \in \{1:n\}}\sum_{r \in \mathcal{M}_i}\left|\mathbf{w}_r(0)^\mathsf{T}\mathbf{x}_p\right|^2 = \mathcal{O}(1). \tag{18}$$

**Condition 3** (Elements of $h_{\mathbf{V}^*,\mathbf{W}(k)}$ are bounded). *For an input sample $\mathbf{z} \in \mathbb{R}^{d_i}$ obtained from $\mathcal{D}$ and for every $k \geq 0$, it follows that with probability at least $1 - \delta$ over the random configuration of $\mathbf{A}$ and input sample $\mathbf{z}$, the following holds $\left|h_{\mathbf{V}^*,\mathbf{W}(k)}(\mathbf{z})[i]\right| = \mathcal{O}(1)$ for each $i \in \{1 : d_o\}$.*

Note that the condition 3 is easily satisfied as each element of network output has a bounded magnitude invariant of $n$ in practice. Further, the conditions 1 and 2 also easily hold by a practical assumption that correlation between a target training sample and a weight column follows the Gaussian variable (i.e., they are independent of each other) if they have no deterministic relation. We formally state it as follows:

**Proposition 3.1.** *(a) Suppose that $|\mathbf{w}_r(0)|$ is invariant of $n$ for any $r \in \{1 : m\}$ (i.e., $|\mathbf{w}_r(0)| = O(1)$). (b) Suppose that given some positive constant $\epsilon$, $|\mathbf{w}_r(0)^\mathsf{T}\mathbf{x}_p| \geq \epsilon$ satisfies for any $r \in \{1 : m\}$ and $p \in \{1 : n\}$ without having randomness. (c) Suppose also that for any $r \in \{1 : m\}$ and $p \in \{1 : n\}$ with having randomness, $\mathbb{P}\left[|\mathbf{w}_r(0)^\mathsf{T}\mathbf{x}_p| \leq x\right] = O(x)$ satisfies for any $x > 0$ (e.g., $\mathbf{w}_r(0)^\top\mathbf{x}_p$ follows the Gaussian distribution). Then, both conditions 1 and 2 hold with $R = O(\frac{\lambda_0}{n^2})$ and $m = O(n^\alpha)$ for some sufficiently large $\alpha$.*

Proposition 3.1 indicates that the conditions 1 and 2 hold even when $\mathbf{W}(0)$ is *partially random* (i.e., only some columns of $\mathbf{W}(0)$ are random and the others have the deterministic relation with training dataset). In addition, we show that the proposed conditions 1 and 2 even hold in the case where $\mathbf{W}(0)$ is completely random, proving its global mildness, as specified in the following remark.

**Remark.** For simplicity, we let $\|\mathbf{x}_j\| = 1$ for all $j \in \{1 : n\}$. Suppose that each element of $\mathbf{W}(0)$ is i.i.d. given as the normal distribution. Then, as Proposition 3.1 holds, with probability at least $1 - \delta$ over $\mathbf{W}(0)$, both the conditions 1 and 2 hold with $R = O(\frac{\lambda_0}{n^2})$ and $m = O(n^\alpha)$ for some sufficiently large constant $\alpha$.

Under the above setup and conditions, then we are now ready to present our theorem, which shows the generalization bound with arbitrary initialization as follows.

**Theorem 3.1** (Generalization Error Bound with Arbitrary Initialization). *Suppose that the conditions 1 ∼ 3 hold, $\|\mathbf{Y}_i\| = O(\sqrt{n})$ for all $i \in \{1 : d_o\}$, $m = \Omega\left(\frac{n^2}{\lambda_0^2 R^2 \delta}\right)$, $m = \Omega\left(\frac{d_o \cdot n^4}{\min(\lambda_0^2, \lambda_0^4)}\right)$, the set $\{\mathbf{x}_j \coloneqq \phi_{\mathbf{V}^*}(\mathbf{z}_j)\}_{j=1}^n$ of $n$ training samples is bounded as $\max_{j \in \{n\}} \|\mathbf{x}_j\| \leq 1$, and $\eta = O(\frac{\lambda_0}{n^2})$. Suppose also that $\lambda_0 = O(n^\gamma) > 0$ with a constant $\gamma \leq 1$ with probability at least $1 - \delta/3$ for $n$ i.i.d. training samples $\{(\mathbf{z}_i, \mathbf{y}_i)\}_{i=1}^n$ from data distribution $\mathcal{D}$. Then, with probability at least $1 - \delta$ over the random initialization of $\mathbf{A}$ and the training samples, it follows that for any $k \geq \Omega(\frac{1}{\eta \lambda_0} \log \frac{n}{\delta})$,*

$$\mathbb{E}_{\mathcal{D}}\left[\frac{1}{2}\|\mathbf{Y} - F(\mathbf{W})\|^2\right] \leq \underbrace{\sum_{i=1}^{d_o} \frac{\left\|\mathbf{H}_i(0)^{-\frac{1}{2}}\big(\mathbf{Y}_i - h_{\mathbf{V}^*, \mathbf{W}(0)}(\mathbf{Z})_i\big)\right\|}{\sqrt{n}}}_{\text{Multi-class Revised CMD}} + \mathcal{O}\left(d_o \sqrt{\frac{\log \frac{n}{\lambda_0^i \delta}}{n}}\right), \quad (19)$$

*where $\mathbf{Y}_i, h_{\mathbf{V}^*, \mathbf{W}(0)}(\mathbf{z})_i \in \mathbb{R}^n$ denote all the collection of labels/outputs of the class $i$, resp.*

Note that the upper bound 1 of condition $\max_{j \in \{n\}} \|\mathbf{x}_j\| \leq 1$ in Theorem 3.1 can be easily extended to the case of any constant other than 1, thereby being satisfied in practice. As the second term in Eq. 19 trivially converges to 0 as $n$ increases, so it can be interpreted that the first term in Eq. 19 represents the generalization error bound. The multi-class revised CMD term in Eq. 19 does not rely on the Gram matrix $\mathbf{H}^\infty$ but on the Gram matrix $\mathbf{H}(0)$, which allows arbitrary initialization. Furthermore, Theorem 3.1 can be thought of as a generalization of Theorem 2.5 in the absence of a feature extractor $\phi_{\mathbf{V}^*}(\cdot)$ since we only have a 2-layer ReLU classifier in this case.

### 3.3 OPENING THE DOOR TO PRACTICE: NTK REGULARIZER IN FINE-TUNING

In this section, we discuss the potential applications based on our revised analysis on generalization error, and as an example, we will introduce how our bound can be utilized in practice. As an observation, we can regard the $k'$-th step parameter $\mathbf{W}(k')$ for any $k' \in \mathbb{N}$ as a new initial parameter $\widetilde{\mathbf{W}}(0)$ so that the parameter $\mathbf{W}(k'+1)$ updated at the next step from $k'$-th step can be viewed as the parameter $\widetilde{\mathbf{W}}(1)$ updated only once from the new initial parameter $\widetilde{\mathbf{W}}(0)$. Since Theorem 3.1 allows arbitrary initialization, this observation provides the following remark.

**Remark.** Suppose that all conditions in Theorem 3.1 for $\mathbf{W}(0)$ hold if $\mathbf{W}(0)$ is replaced with $\mathbf{W}(k)$ at any step $k$. Then, the generalization error bound can be again characterized as

$$\frac{1}{\sqrt{n}} \sum_{i=1}^{d_o} \left\|\mathbf{H}_i(k)^{-1/2}\big(\mathbf{Y}_i - h_{\mathbf{V}^*, \mathbf{W}(k)}(\mathbf{Z})_i\big)\right\|. \quad (20)$$

where $\mathbf{H}_i(k)$ is defined in Eq. 16. By the above observation and remark, the multi-class revised CMD in Eq. 20 could further be thought of as the generalization error bound of some fine-tuned networks. This fact motivates the following NTK regularizer,

$$\mathcal{R}_{\text{NTK}}(\mathbf{W}) = \frac{1}{\sqrt{n}} \sum_{i=1}^{d_o} \left\|\mathbf{H}_i(\mathbf{W})^{-1/2}\big(\mathbf{Y}_i - h_{\mathbf{V}^*, \mathbf{W}}(\mathbf{Z})_i\big)\right\|, \quad (21)$$

and training with the regularization loss can play a crucial role in directly reducing the generalization error. Therefore, we propose to solve the following optimization problem in fine-tuning: $\min_{\mathbf{W}} \{\mathcal{L}(h_{\mathbf{V}^*, \mathbf{W}}(\mathbf{z}), \mathbf{y}) + \mu \mathcal{R}_{\text{NTK}}(\mathbf{W})\}$ where $\mu$ represents the strength of the regularizer.

In fact, the proposed NTK regularizer in Eq. 21 requires the computations of inverse Gram matrix $\mathbf{H}_i(k)^{-1/2}$ for each class $i \in \{1 : d_o\}$, which makes network training computationally heavy in practice. To bypass this issue, we suggest to use the fixed Gram matrix $\mathbf{H}_i(0)$ computed on the initial parameter $\mathbf{W}(0)$ for all $i$ (in fine-tuning regime, the initial parameter $\mathbf{W}(0)$ boils down to some pre-trained parameter $\mathbf{W}^*$). The intuition behind this is as follows: the generalization error bound depends on the multi-class revised CMD as in Theorem 2.5 for arbitrary initialization. In the over-parametrization regime, since the model parameter will remain close to its initial point during training, the Gram matrix $\mathbf{H}_i(k)$, depending on the $k$-th model parameter $\mathbf{W}(k)$, will also stay close to its initial Gram matrix $\mathbf{H}_i(0)$ for all $i$. Though the Gram matrix $\mathbf{H}_i(k)$ is fixed to $\mathbf{H}_i(0)$ in NTK regularizer in Eq. 21, the gradient-based training on regularized loss is still possible since the gradient with respect to $\mathbf{W}$ will be backpropagated through $h_{\mathbf{V}^*, \mathbf{W}}$ in $\mathcal{R}_{\text{NTK}}(\mathbf{W})$.

Table 1: Comparisons of generalization error bound: Theorem 2.2 (baseline) vs. Theorem 2.5 (ours). Note that *the error term $\mathcal{E}$ is absent in Theorem 2.5 of our revised analysis*. Our revised theory *removing the error term* achieves significantly lower generalization error bound.

| Dataset | Error Term $\mathcal{E}$ in Eq. 7 | Original CMD | Revised CMD (Ours) | Original Gen. Err. Bound | Revised Gen. Err. Bound (Ours) |
|---|---|---|---|---|---|
| MNIST | $7 \times 10^3$ | 0.5998 | **0.5997** | $\mathcal{O}(10^3)$ | $\mathcal{O}(\mathbf{10^{-1}})$ |
| FashionMNIST | $1 \times 10^5$ | **0.2617** | 0.2618 | $\mathcal{O}(10^5)$ | $\mathcal{O}(\mathbf{10^{-1}})$ |
| CIFAR-10 | $4 \times 10^3$ | 2.0605 | **2.0604** | $\mathcal{O}(10^3)$ | $\mathcal{O}(\mathbf{1})$ |

Lastly, note that our NTK regularizer is expected to exhibit its greatest effect in boosting generalization given a considerably lack of data, rather than in cases where a sufficient number of samples are available to achieve plausible performance. Also, the inverse Gram matrix involves $\mathcal{O}(n^3)$ computations w.r.t. sample size $n$, we mainly focus on the lack-of-data scenario for evaluating NTK regularizer.

## 4 NUMERICAL SIMULATIONS

The primary goal in experiments is to verify (i) the tighter generalization error bound of our revised analysis in Section 2 and (ii) whether the generalization is indeed improved via NTK regularizer, which will corroborate our theory in Section 3.

### 4.1 THEORY VALIDATION: COMPARIONS OF GENERALIZATION ERROR BOUNDS

In order to compare the generalization error bounds between Theorem 2.2 and Theorem 2.5, we consider 2-layer ReLU networks with the width $m = 10000$. Since the baseline theory includes an error term $\mathcal{E}$ which is absent in our revised bound, the key points in comparing the generalization error bound is (i) how much the baseline CMD term differs from the revised CMD term, and (ii) the scale of the error term. Toward this, we consider three benchmark datasets: (i) MNIST, (ii) FashionMNIST, and (iii) CIFAR-10. While our theory can allow the multi-dimensional outputs as in Theorem 3.1, the baseline error bound in Theorem 2.2 could only guarantee the regression or binary classification (refer to Corollary 5.2 in Arora et al. (2019a)), i.e., single output case. Thus, we randomly pick two classes for each dataset. The revised CMD term depends on the initial network output $\mathbf{u}(0)$, thus we initialize $\mathbf{W}(0)$ with the practical Kaiming normal distribution (He et al., 2015) whose scaling does not rely on the sample size $n$ at all, which violates the conditions of Theorem 2.2.

Table 1 illustrates the direct comparison of generalization error bound. Note first that the revised CMD does not significantly differ from the original CMD. Although the revised CMD is slightly larger than the original CMD in FashionMNIST dataset, the difference is only on the scale of $10^{-3}$. To compute the scale of the error term $\mathcal{E}$ in Eq. 7, which is the second most important factor in the comparison of generalization errors, we set the failure probability $\delta = 0.01$ (larger value is also fine). Note that the error terms $\mathcal{E}$ have the scale of $10^3 \sim 10^5$ while the CMD terms have values only about 0.6, 0.26, and 2.06 for each dataset as can be seen in Table 1. Hence, our revised analysis *removing the error terms* could significantly improve the existing generalization error bound. It is important to note that the results in Table 1 are not limited to solely to the width $m = 10000$, since our findings hold true across a broad range of width $m$ from $10^2$ to $10^4$, and we provide the results in Appendix A.

### 4.2 VERIFICATION OF MULTI-CLASS REVISED CMD VIA NTK REGULARIZER

In order to verify our theory (Theorem 3.1) on the arbitrary initialization, we use the NTK regularizer proposed in Eq. 21. Toward this, we fine-tune pre-trained models given a limited number of samples, which closely mirrors the typical scenario in medical applications. Thus, we consider the skin cancer classification for our experiments. The details on experimental settings are provided in Appendix.

**Model.** We use pre-trained ResNet-18 (He et al., 2016) on the ImageNet (Deng et al., 2009), which is publicly available from popular deep learning libraries (Abadi et al., 2016; Paszke et al., 2019). As

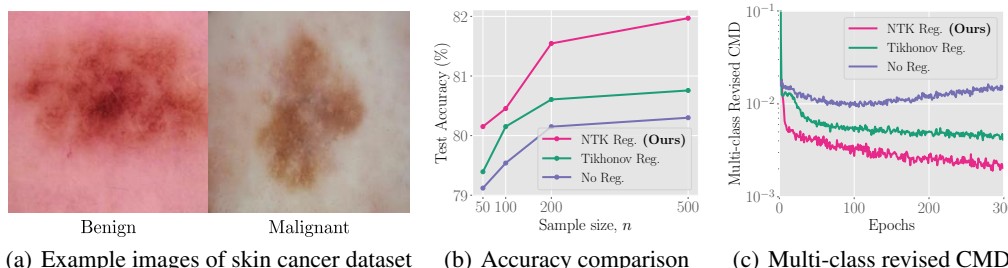

(a) Example images of skin cancer dataset     (b) Accuracy comparison     (c) Multi-class revised CMD

Figure 2: **(a)** Example images of skin cancer dataset, **(b)** the results on skin cancer varying the number of training samples. **(c)** comparison of multi-class revised CMD among different methods.

suggested in our theory, we replace the classifier of ResNet-18 with a 2-layer ReLU network, and the parameter $\mathbf{A}$ of second layer of the classifier is initialized and fixed according to Section 3.1. The parameter $\mathbf{V}^*$ of the feature extractor is frozen to those of the pre-trained model (one of conventional fine-tuning strategy), while only the parameter $\mathbf{W}(0)$ of the first layer of the classifier is updated.

**Dataset.** The skin cancer classification has been considered as one of popular medical applications in many literature (Esteva et al., 2017; Wu et al., 2022; Bello et al., 2024). The goal of this task is to classify types of skin cancer for given images into: (i) benign or (ii) malignant. In this experiment, we collect RGB images of size $224 \times 224$ of combined dataset, which consists of HAM10000 (Tschandl et al., 2018) and International Skin Imaging Collaboration (ISIC 2020). The dataset is splitted into $2077/560/660$ images for train/valid/test respectively and we provide example images of the dataset in Fig. 2(a) for better understanding. The detailed information of dataset is provided in Appendix.

**Baselines.** To validate the efficacy of NTK regularizer, we consider two baselines: (i) no regularizer (regular fine-tuning) and (ii) Tikhonov regularizer corresponding to the case of $\mathbf{H}_i(k)$ being the identity matrix $\mathbf{I}$ in Eq. 21, i.e., $\mathcal{R}_{\text{Tikhonov}}(\mathbf{W}) = \frac{1}{\sqrt{n}} \sum_{i=1}^{d_o} \|\mathbf{Y}_i - h_{\mathbf{V}^*, \mathbf{W}}(\mathbf{Z})_i\|$. The reason for considering the Tikhonov regularizer is to examine the role of the Gram matrix in $\mathcal{R}_{\text{NTK}}$.

**NTK regularizer works in practice.** We consider small amount of training dataset to simulate a limited-number-of-sample scenario. Toward this, we randomly choose $\{50, 100, 200, 500\}$ samples from training dataset, on which we fine-tune the pre-trained ResNet-18. As depicted in Fig. 2(b), NTK regularizer indeed improves the generalization upon regular fine-tuning. Note that the advantage of NTK regularizer over the Tikhonov regularizer clearly can be clearly observed, which indicates that the Gram matrix plays an important role in model generalization. In addition, we also compare the multi-class revised CMD term as illustrated in Fig. 2(c). An interesting observation is that the multi-class revised CMD decreases as the model generalization improves observed in Fig 2(b). This suggests that directly reducing the multi-class revised CMD can potentially enhance the generalization, which demonstrates the validity of our proposed NTK regularizer.

## 5   CONCLUSION

In this study, we revised the existing NTK-based theory of optimization and generalization for overparametrized neural networks. Our revised analysis successfully remove the unreasonable assumption on the initialization and provide tighter bound for generalization error. Going further, we extended our revised analysis that allow for arbitrary initialization and multi-dimensional outputs. By extending NTK theory to a network with arbitrary initialization, we were able to propose the concept of NTK regularzer, which was previously unattainable, and validate its effectiveness. *The most promising aspect of this study is that it enables the application of NTK theory to pre-trained networks.* This extension of NTK theory is expected to be applicable to various practical scenarios that require predicting the performance of pre-trained networks, such as fine-tuning, domain adaptation, out-of-distribution detection, and more. We empirically validated that our revised analysis indeed achieve significantly lower generalization error bound and also showed our NTK regularizer to be effective in fine-tuning, demonstrating that NTK theory provides a connection to real-world applications.

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
