# OpenReview forum: "Revised NTK Analysis of Optimization and Generalization with Its Extensions to Arbitrary Initialization"
_ICLR.cc/2025/Conference — ICLR 2025 Conference Withdrawn Submission_

### Official Review · Reviewer_jd1v · 2024-10-16

**Soundness:** 3
**Presentation:** 2
**Contribution:** 2
**Rating:** 5
**Confidence:** 3

**Summary:**

This paper mainly builds on the results in  Arora et al. 2019, focusing on the optimization and generalization of over-parameterized neural networks. The paper addresses a limitation in NTK-based analysis in  Arora et al. 2019, which requires the scaling of initial parameters to decrease with respect to the sample size, a condition that contradicts practical initialization schemes. To resolve this issue, the authors try to removes the dependency of initialization on sample size, and extend to applying the method in real practice.

**Strengths:**

This paper extends the work of Arora et al. 2019, removing the dependency of initialization on sample size. The authors provide a practical approach that aligns with real-world initialization schemes. This contribution enhances the applicability of NTK theory in real-world scenarios, and can be extended to future research.

**Weaknesses:**

The generalization upper bound in Theorem 2.5, approximated as $O(1)$, may not be sufficiently tight compared to more recent works that provide tighter bounds, such as Suh et al. (2021) and Hu et al. (2021), which suggest a bound of $O(n^{-\frac{d+1}{2d+1}})$. A bound that does not decrease with sample size $n$ raises concerns about its effectiveness. At the same time, the characterization of generalization capability in this paper is precisely based on this upper bound, which makes me feel that it is not solid enough.​


## References

[Suh et al., 2021] Suh, N., Ko, H., and Huo, X. (2021). "A non-parametric regression viewpoint: Generalization of overparametrized deep ReLU network under noisy observations." In _International Conference on Learning Representations_.

[Hu et al., 2021] Hu, T., Wang, W., Lin, C., and Cheng, G. (2021). "Regularization matters: A nonparametric perspective on overparametrized neural networks." In _International Conference on Artificial Intelligence and Statistics_, pp. 829–837. PMLR.

**Questions:**

I note that the generalization upper bound in Thm 2.5 is

$$O\left( \sqrt{ \frac{2(y - u(0))^T (H^\infty)^{-1} (y-u(0))}{n}} + \sqrt{\frac{\log {\frac{n}{\lambda_0 \delta}}}{n}}\right),$$

which is about $O(1)$ since $y - u(0)$ is a $n$-length vector. However, during these years, I have seen lots of recent work which provide tighter upper bounds for generalization error of networks based on NTK (e.g., $O(n^{-\frac{d+1}{2d+1}})$,[Suh et al., 2021], [Hu et al., 2021],), which shows that maybe $O(1)$ is not tight enough.  At the very least, an upper bound on the generalization error that does not decrease with increasing sample size n is hard to consider as tight.  I think maybe it is a better choice to compare the result with theirs.

In Table 1, I see that this paper compared "Original CMD" with "Revised CMD", which is generally the same (e.g., 0.5998 with 0.5997). Table 1 also compares the generalization upper bound, which is common due to the removal of the initialization scale factor. So I am slightly confused by the meaning of Table 1. In Figure 2, I think the experiment should represent more details to make Figure 2(b) reliable (e.g., the stopping time of training).

Additionally, I don't understand the y-axis of Figure 2(c). I thought CMD is time-invariant and should not change with training, so I am confused about what the y-axis of Figure 2(c) represents. I would appreciate it if you could correct my misunderstanding.

---

### Official Review · Reviewer_MMW5 · 2024-10-31

**Soundness:** 2
**Presentation:** 1
**Contribution:** 2
**Rating:** 1
**Confidence:** 5

**Summary:**

This paper studies the optimization and generalization problems of over-parameterized neural networks, proposes a revised NTK framework that eliminates the dependency between initialization scale and sample size, and verifies its experimental results on benchmark datasets.

**Strengths:**

The revised NTK framework proposed in this paper solves the dependency problem between the NTK initialization scale and the sample size. In addition, the numerical experiments in this paper support improving the proposed regularization method in improving generalization performance.

**Weaknesses:**

The writing of the paper is not clear, especially in Appendix B, where the proof structure and formula numbering are very confusing and difficult for readers to understand. In addition, the lack of proof of Theorem 3.1 in the appendix makes it impossible to confirm the correctness of the theoretical results of the paper. Overall, the paper is not yet in a submittable state. I suggest authors complete any unfinished parts and thoroughly reorganize the paper to improve its clarity and readability before submitting it to a future conference.

**Questions:**

Could the authors provide a complete proof of Theorem 3.1?

Could the authors clarify the formula numbering and proof structure in Appendix B in a future version?

---

### Official Review · Reviewer_HJkp · 2024-11-02

**Soundness:** 1
**Presentation:** 2
**Contribution:** 1
**Rating:** 3
**Confidence:** 5

**Summary:**

The paper revises the Neural Tangent Kernel (NTK) in analyzing the generalization performance of neural networks at the infinite width limit, focusing particularly on the impact of the scale $\kappa$ of initialization over the bounds for optimization and generalization. Showing that the error bounds in the previous works  Arora et al. (2019a;b) actually does not hold in the case $\kappa = o(1)$ which is necessary for the bound to be non-vanishing. Then, this paper revises the previous result, establishing optimization and generalization bounds independent of $\kappa$. Basing on the new error bounds, the paper discusses the generalization error bounds with arbitrary initialization. Finally, numerical simulations are provided to verify the theory.

**Strengths:**

- The orginaztion of the paper is clear and easy to read.
- Revising the impact of initialization to the generalization error bound is an interesting problem and extends the existing theory of neural tangent kernel.

**Weaknesses:**

### Correctness

One major problem is the correctness of the results in this paper.
In the proof of Lemma 4, the paper applies the Markov’s inequality for $\mathbf{a}$ with fixed $\check{W} \in \Gamma$ to prove Eq. (41), and then use (41) for $W(k)$ subsequently, which is shown to lie in $\Gamma$. However, since $W(k)$ is random and dependent on $\mathbf{a}$, there is no guarantee that Eq. (41) holds for $\check{W}$ replaced with $W(k)$. To make this approach work, a uniform version of Eq. (41) over $\Gamma$ is needed. Given that Lemma 4 is not well justified, the main results in this paper are not well-supported.

### Novelty

The results regarding $\kappa = \Theta(1)$ are not novel and are slight modifications of the existing results. While the proof seems to be long, they are mostly minor revisions of the existing proofs.
Also, this paper considers the simplest setting of a two-layer ReLU network with only the first layer trainable. However, as existing NTK theory (for example, [1]) can deal with more general settings such as multi-layer networks, extensions to these settings should also be considered.

Moreover, regarding the generalization error bounds, this paper misses some related literature studying the NTK regime in terms of kernel regression [2,3], where sharper bounds are established and also do not depend on the scale of initialization (as the NTK in this setting does not depend on the scale).
I think a more detailed comparison with the existing works should be provided.


### Minor

Some notations are used without definition and are not consistent. For example, $H^*$ is used  in Section B.2 but defined in Section C, which seems to refer to $H^\infty$ in the main text.

When reviewing this paper, I also find a very recent paper concerning the impact of initialization [4]. A comparison with this paralleling work would be benefitial to the readers.

### References

[1] Zeyuan Allen-Zhu, Yuanzhi Li, and Zhao Song. A convergence theory for deep learning via over-parameterization. In International Conference on Machine Learning, pages 242–252. PMLR, 2019.

[2] Namjoon Suh, Hyunouk Ko, and Xiaoming Huo. A non-parametric regression viewpoint: Generalization of overparametrized deep relu network under noisy observations. In International Conference on Learning Representations, 2021.

[3] Tianyang Hu, Wenjia Wang, Cong Lin, and Guang Cheng. Regularization matters: A non-parametric perspective on overparametrized neural network. In International Conference on Artificial Intelligence and Statistics, pages 829–837. PMLR, 2021.

[4] On the Impacts of the Random Initialization in the Neural Tangent Kernel Theory, Guhan Chen, Yicheng Li, Qian Lin. https://arxiv.org/abs/2410.05626

**Questions:**

I would like the authors to response to the weakness section.

---

### Official Review · Reviewer_yPNZ · 2024-11-04

**Soundness:** 3
**Presentation:** 1
**Contribution:** 3
**Rating:** 6
**Confidence:** 3

**Summary:**

This work focus on two problems: under what conditions the training error bound and the generalization error bound are obtained and go to zero in the NTK setting. The paper disproves the work of Arora and then provides proofs for better results, particularly for the case the initial values of training weights do not decrease with the sample size.

**Strengths:**

The work provides a list of theoretical results. I can only check some of them and believe they are correct.
It proposes a more realistic setting for the NTK.

**Weaknesses:**

The crucial condition for Theorem 2.3 hold is $\kappa = O(n^{\alpha})$ with $\alpha <0$, that only appears its proof in the Appendix. Based on the condition, the considered setting is very specific, that is the initial weights depend on $n$.  Let $n$ go to infinity, the $\|\mathbf{y}\|^2$ grows as $n$, the distance from initial weights to its desired destination  increases as $n$ increases. So intuitively what is surprise from the result: the training weight cannot converge to the weight in which model has zero loss?

It is often that the paper states some theoretical results depending on some conditions, then those conditions happen under assuming another set of conditions? Could the author state everything together so that readers can check them all? For example,  Proposition 3.1 states condition 1 and 2 hold, when requiring other conditions. In Proposition 3.1, $\epsilon$ appears, does $\epsilon$ affect the conditions on $m, n$ and other parameters in the proofs of other theoretical results?


The paper is very long, full of technical, not well-organized. There is no explanation for intuition behind the proofs and the proof's structure. For example, equation (24) appears at the beginning of the Appendix's proof refers to equation (138), which is at 20 pages later.  It is not easy for reader to read 50 pages of proof with that presentation. It is unfair to reject the paper just due to poor presentation, but it makes reader hard to verify technical results to be certain that they are all correct.

In short, could the authors make a table comparing this work and the work of Arora in term of conditions and results and then highlight the technical advancement of this work, also give a discussion about all related parameters in one. Since for this kind of theoretical work, the most difficult task is to be certain that all conditions do not conflict each other and cover a wide range of cases.

Minors
1. Unconventional notation $\{i\}$ for the set $\{1,2,\ldots, i\}$
2. Math notations are not consistent, not defined, $\Omega(1), O(n), \Omega(n), \Theta(n)$, $o(1)$ with respect to $n$.
3. I could not find the proof of Proposition 3.1???
4. Line 375, how large is $\alpha$ acceptable, since $m$ depends on $n$ in other results?

**Questions:**

Please look at the weaknesses.

---

### Note · Authors · 2024-11-26

**Comment:**

Dear AC and Reviewers,

Thank you for your valuable feedback and insightful comments on our submission. We deeply appreciate the time and effort you dedicated to reviewing our work.

After carefully considering the reviewers' comments, we have identified areas of the work that require further refinement and deeper exploration. While we remain confident in the potential of our approach, we believe that addressing these points thoroughly will result in a stronger and more robust contribution to the field.

In light of this, we have decided to withdraw our submission at this time to allow us the opportunity to revisit and strengthen the work. This decision reflects our commitment to ensuring the highest quality in our research.

**Withdrawal Confirmation:**

I have read and agree with the venue's withdrawal policy on behalf of myself and my co-authors.